# The influence of Holliday junction sequence and dynamics on DNA crystal self-assembly

Chad R. Simmons[1,5], Tara MacCulloch[1,2,5], Miroslav Krepl [3,4], Michael Matthies[1], Alex Buchberger[1,2], Ilyssa Crawford[1,2], Jiří Šponer [3,4], Petr Šulc [1,2], Nicholas Stephanopoulos [1,2✉] & Hao Yan [1,2✉]

The programmable synthesis of rationally engineered crystal architectures for the precise arrangement of molecular species is a foundational goal in nanotechnology, and DNA has become one of the most prominent molecules for the construction of these materials. In particular, branched DNA junctions have been used as the central building block for the assembly of 3D lattices. Here, crystallography is used to probe the effect of all 36 immobile Holliday junction sequences on self-assembling DNA crystals. Contrary to the established paradigm in the field, most junctions yield crystals, with some enhancing the resolution or resulting in unique crystal symmetries. Unexpectedly, even the sequence adjacent to the junction has a significant effect on the crystal assemblies. Six of the immobile junction sequences are completely resistant to crystallization and thus deemed "fatal," and molecular dynamics simulations reveal that these junctions invariably lack two discrete ion binding sites that are pivotal for crystal formation. The structures and dynamics detailed here could be used to inform future designs of both crystals and DNA nanostructures more broadly, and have potential implications for the molecular engineering of applied nanoelectronics, nano-photonics, and catalysis within the crystalline context.

---

[1] Biodesign Center for Molecular Design and Biomimetics, Arizona State University, 1001S. McAllister Ave, Tempe, AZ 85287, USA. [2] School of Molecular Sciences, Arizona State University, Tempe, AZ 85287, USA. [3] Institute of Biophysics of the Czech Academy of Sciences, Královopolská 135, 612 65, Brno, Czech Republic. [4] Regional Centre of Advanced Technologies and Materials, Czech Advanced Technology and Research Institute (CATRIN), Palacky University Olomouc, Slechtitelu 241/27,783 71, Olomouc, Czech Republic. [5] These authors contributed equally: Chad R. Simmons, Tara MacCulloch. ✉email: nstepha1@asu.edu; hao.yan@asu.edu

The fabrication of highly customizable 3D DNA-based architectures for the precise organization of nanoscale materials was originally conceptualized by Seeman in 1982[1]. A variety of methodologies that allow for programmable self-assembly have been developed, including the attachment of DNA linkers to the nanoparticle (NP) surfaces[2,3] for the construction of 3D lattices with user-defined colloidal NP crystal configurations[4–8]. With the advent of DNA origami[9], 3D superlattices of nanoparticle cluster shapes have been described[10,11] along with 3D origami lattices displaying customizable geometry to host guest species[12]. Rationally designed crystals based upon the "tensegrity" motif[13] which self-assemble with designed four-arm junction crossover points and "sticky end" cohesion have also been shown to be capable of being converted into nanodevices[14]. Recently, several unique motifs with distinctive crystal symmetries, improved resolution, and in one case, a unique junction nucleotide sequence, have been reported[15–18].

The application of four-way junctions for 3D DNA crystals was inspired by genetic recombination, whereby an unstable branched intermediate termed a Holliday junction (HJ) is created and subsequently undergoes a dynamic reconfiguration to facilitate recombination during cell division[19]. Holliday junctions have been extensively structurally characterized[20–29], and emerged as a key motif for rationally designed nanoscale assemblies and devices in structural DNA nanotechnology[1,30] Naturally occurring HJs can "slide" and change the length of their arms[31,32], a process known as branch migration[33]. Introducing asymmetric sequences at the branching point, however, effectively immobilizes the junction and allows its use in constructing well-defined nanostructures[34]. Although a variety of multi-branch junctions have been employed[35–41], the four-arm HJ remains the most popular.

Theoretically, there are 36 base pair combinations of immobile sequences. "J1" was the earliest to be designed[42,43], and has been used nearly exclusively in the construction of self-assembled 3D crystals[13,15–17] with a single exception where "J10" was used[18]. Some early work explored the sequence influence on packing of stacked-X junctions using theoretical[44,45] or experimental methods[46,47], but to our knowledge, no systematic study of the immobile HJs has been performed. In addition, the relationship between ion concentration and the transition of the HJ from the open to the stacked conformation is well known[20,25,48–51]; however, studies of sequence effects on the ability to bind ions are limited, making the elucidation of the structural parameters that influence crystallization and symmetry a desirable route for exploration.

In this work, we exhaustively probe the ability of all 36 immobile HJ sequences to form DNA crystals in two different designed systems. Our work revealed that a large majority of immobile HJs enable crystallization, with some yielding higher resolution structures and a variety of symmetries. The symmetry is also highly sensitive to the sequence of the junction arms (stems) further away from the branching point, as we demonstrate using scrambled stem sequences. Unique ion binding sites were also observed at two conserved positions within the structures. By performing molecular dynamics (MD) simulations of all 36 HJs in solution, we show that these sites are pivotal for crystallization, with the universally non-crystallizing "fatal" junctions showing zero ability to bind ions in this fashion. Although limited by sampling, the modeling used in this work provides a reliable picture of sequence-dependent flexibility and solvent effects of all 36 immobile HJs as they relate to self-assembled DNA crystal lattices. Overall, the work provides a rigorous and complete description of how different immobile HJ sequences, flanking sequence modifications, and the ability of the junction to capture ions profoundly affect the rational design of self-assembled systems that span a multitude of designer DNA architectures.

## Results and discussion

Three separate self-assembling DNA crystal systems are described in this report: the "4 × 5"[15] and "4 × 6"[16] designs (collectively referred to as the $4 \times N$ systems), and a third construct with a "scrambled" sequence variant of the 4 × 6 lattices. Self-assembly was mediated by three constituent oligonucleotides (Fig. 1a): (S1) containing four sequence repeats of either 5 or 6 bases; (S2) composed of 21 bases containing complementary regions to both (S1); and (S3) which forms the second junction crossover (Fig. 1b). Each asymmetric unit could be defined as either a HJ containing 10 and 11 bp on each arm, or as a 21-bp linear duplex (Supplementary Fig. 1). Both versions were solved in parallel, yielding a total of 134 crystal structures (Supplementary Table 1 for data collection and refinement statistics). The crystal lattices contain continuous arrays comprised of a series of crystal "blocks" that self-assemble into a series of 21-bp duplexes tethered by the 4×N strand. The HJ serves as the fundamental component at the core of each unit, and the ultimate assembly of the full lattice is facilitated by the complementary 2-base sticky ends that tail each duplex. The "flanking" sequences in the original 4 × 6 were also altered on opposing sides to identify any role the downstream sequence might play in crystallization behavior, independently or in conjunction with the HJ sequence itself.

Here, we probed junction sequence space by creating a panel of 36 immobile HJs (Fig. 1f) in each system while considering a single fixed isomer to determine if junctions other than J1 were capable of crystallizing and potentially yielding improved resolution, and to identify those that proved fatal. The sequences were explicitly defined on each constituent junction strand (all component oligonucleotide sequences can be found in Supplementary Data 1 and Supplementary Table 2), or when defined as a 21-bp duplex (Fig. 1, Supplementary Fig. 2, and Supplementary Fig. 3). To definitively confirm a junction as fatal, rigorous screening (Supplementary Table 3) was performed to conclusively classify each sequence (See "Methods" in the Supplementary Information for technical details). The resulting structures contained three different symmetries ($P3_221$, $P3_2$, and $R3$) that are described in detail below (Fig. 1e, Supplementary Fig. 4).

### Junctions enhance resolution and influence crystal symmetry (4 × 5 motif).

The original 4 × 5 structure containing the J1 junction sequence resulted in a structurally strained array, presumably due to the underwinding to the central component strand[15]. The layered motif provided a well-defined scaffold at 3.1 Å resolution, but the aperiodicity of the cavities and corresponding volumes would be inadequate for scaffolding guest materials. We hypothesized that it might be possible to introduce other unique immobile sequences that could provide slight perturbations in the junction angle, to potentially reduce the strain in the original system and yield a "relaxed," periodic lattice.

The 4 × 5 system robustly crystallized with 75% of the junctions, and we obtained crystals in all but 9 of 36 sequences (Supplementary Fig. 5 and Supplementary Table 4). Junctions J2 and J30 did crystallize, but were of inadequate quality for structure solution, and therefore classified as fatal. Of the resulting structures, 18 exhibited $P3_2$ symmetry with average cell dimensions $a = b = 68.85$ Å $c = 60.09$ Å, with only nine retaining the original $P3_221$ symmetry with average cell edges $a = b = 68.17$ Å $c = 60.60$ Å (Supplementary Table 5). Although the cell parameters between the two symmetries were virtually indistinguishable (Fig. 2a, Supplementary Fig. 6), the differences between the periodicity of the lattices is dramatic, with vastly different cavity sizes (Fig. 2b, c). While the highest resolution achieved for the J1 system was 3.1 Å, in roughly half of the crystals measured the resolutions were 3.05 Å or better, and as high as 2.9 Å

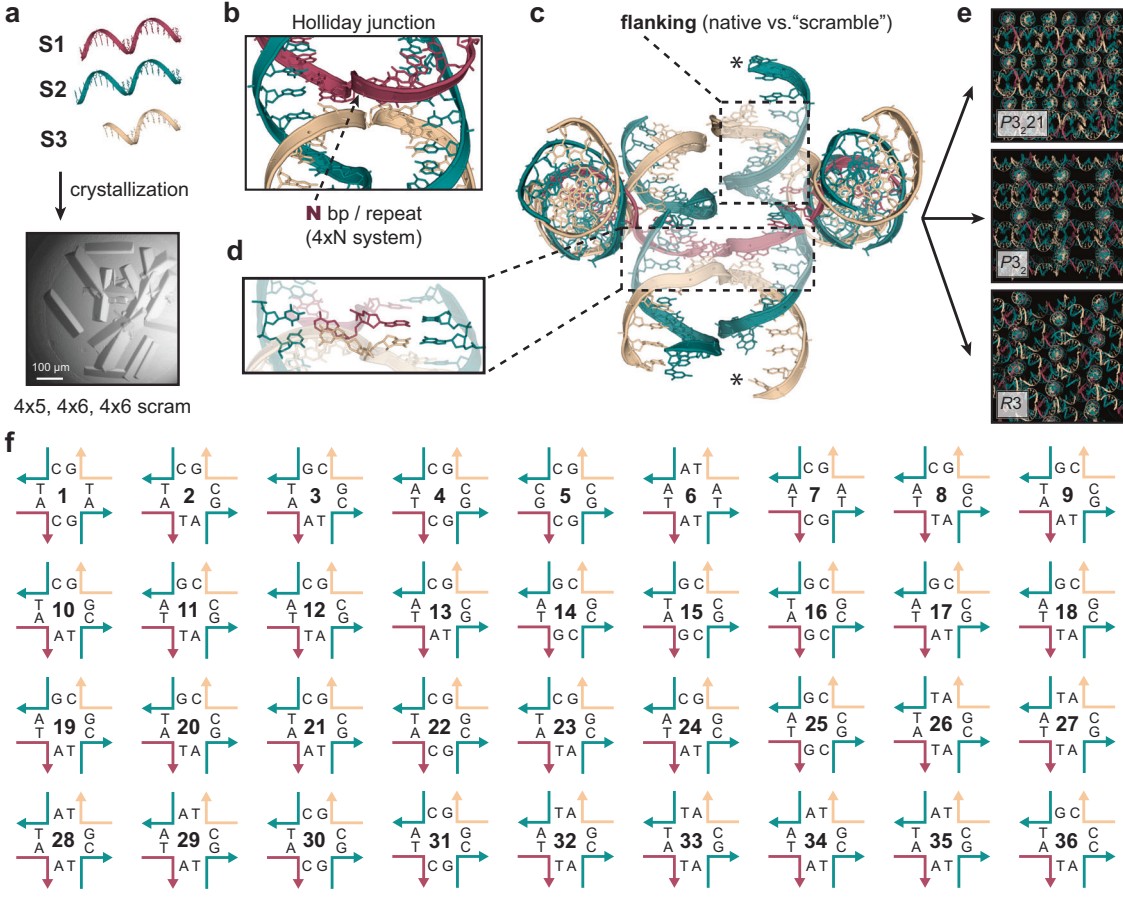

**Fig. 1 Schematic of the composition of the Holliday junction which is required for the self-assembly of 3D DNA lattices. a** Three oligonucleotides mediate the crystallization of two self-assembling motifs ($4 \times 5$ and $4 \times 6$) along with a "scrambled" sequence variant. A representative example of a 3D crystal is shown. **b** The structure of the Holliday junction is the key building block for assembly, and contains four arms using two oligonucleotides (S1; red) and (S3; tan) serving as crossover strands, with S2 (green) serving as a third "linear" complementary strand on each side. The complementary region of S1 contained either five or six bases (N bp) on each arm before each crossover at which point an identical sequence repeats in each consecutive arm for a total of four times ($4 \times N$) before beginning the series again ($4 \times 5$ or $4 \times 6$). S1 subsequently serves as the scaffolding strand for the entire lattice. **c** The central building "block" that facilitates the 3D assembly. S1 tethers four 21-bp duplexes with the Holliday junction (boxed; translucent) at the core of the structure. The linear 21 base ssDNA oligonucleotide (S2) comprises one half of each duplex with the second crossover strand (S3) flanking each end (boxed). Each duplex is tailed by 2 bp complementary "sticky ends" (asterisks) which cohere to form continuous 3D arrays. **d** Representative bases where sequence asymmetry was imposed to prevent "sliding" of the strands to create 36 immobilized junctions. **e** Three unique symmetries ($P3_221$, $P3_2$, and $R3$) are dictated by the $4 \times N$ scaffolding strand working in concert with the sequence at each immobile junction. **f** The 36 immobile junction sequences represented in an open Holliday junction format with each strand colored in accordance with (**b**). Nucleotides on each corresponding strand are indicated with the sequence positions on each component oligonucleotide corresponding to the colored scheme in (**d**).

(J6) and 2.75 Å (J19), for the $P3_2$ and $P3_221$ symmetries, respectively (Supplementary Table 6).

The cavity volumes of the $P3_221$ lattices were calculated based upon the 3-nm edges of the pores along the three-fold symmetry axis of the crystal with a triangular prism height of 6.1 nm, corresponding to the average c-axis of the unit cell (Supplementary Fig. 7). The aperiodic lattice contained cavities with widths of 1.0 and 1.7 nm at alternating intervals, and yielded exceedingly small pore volumes of ~24 nm³ (Fig. 2b; Supplementary Fig. 7a). The $P3_2$ channels were treated as a hexagonal prism with 6.4 nm along each edge and a height measured from the top to the bottom of each block along with the average c-axis = 6.0 nm (Supplementary Fig. 7b), with a resulting volume of ~639 nm³, a nearly 27-fold increase in cavity size relative to the $P3_221$ lattices. The junction PDB coordinates for each symmetry type were grouped, and their angles were analyzed in DSSR[52]. The mean angles across all junctions within the groups were 56.05° ($\sigma = 1.63$) and 56.59° ($\sigma = 1.50$) for the $P3_2$ and $P3_221$, respectively (Supplementary Table 7). We hypothesize that even

a small difference in junction angles could have a significant global effect on the assembly of the lattice, thereby eliminating the strain in the $P3_221$ lattices.

**Junction sequences can dramatically alter crystal symmetry ($4 \times 6$ motif).** In parallel, we employed the previously reported J1 $4 \times 6$ system (3.05 Å), with $P3_2$ symmetry[16]. All 36 junction sequences were screened, and consistently fatal junctions were identified. Seventeen of the 36 junctions successfully crystallized, a 47% success rate (Supplementary Fig. 8 and Supplementary Table 8). Unlike the $4 \times 5$ motif, nearly all of the $4 \times 6$ junction sequences had a strong preference for crystallization in buffers containing ≥2.0 M salts (e.g., LiCl, Li₂SO₄, KCl, and NaCl), with the majority preferring slightly basic pH conditions in cacodylic acid (Supplementary Table 9). We saw no appreciable improvements in resolution; however, in five cases (J4, 5, 31, 33, and 36), the junction altered the symmetry from trigonal ($P3_2$) to rhombohedral ($R3$). Furthermore, J4 and J36

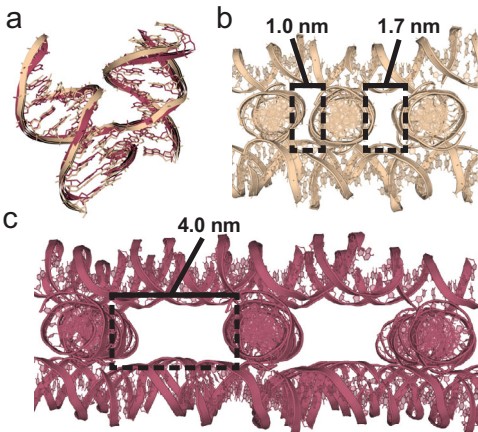

**Fig. 2 Discrete differences in junction angles determine global lattice symmetry. a** Superimposed structures of the J5 (tan) and J3 (red) 4 × 5 junctions containing $P3_221$ and $P3_2$ symmetries, respectively. The junction alignment had a global RMSD value of 1.34, with calculated interduplex angles of 58.18° and 55.20°, respectively. No significantly obvious visual differences are apparent; however, the resulting global influence that an even modest difference in angle can have on overall packing is evident in (**b**, **c**). **b** Snapshot of the full J5 $P3_221$ (4 × 5) lattice containing an aperiodic array of cavities which would not be amenable for scaffolding of guest molecules of any appreciable size. The two uniquely sized cavities are shown with black boxes. The widths for each respective cavity are indicated. Each cavity spans the length of a cross-section of a duplex (~2.0 nm). Alternative views of the lattice including measurements in each orientation are included in Supplementary Fig. 8. **c** Snapshot of the full J3 $P3_2$ (4 × 5) lattice which reveals dramatically different arrays of large periodic cavities compared to its J5 counterpart in (**b**). A single cavity is highlighted with a black box with a width of 4.0 nm and also spanning the length of a cross-section of a duplex (~2.0 nm). Alternative views of the lattice including measurements in each orientation are included in Supplementary Fig. 8.

crystallized exclusively in $R3$, but J5, 31, 33 exhibited the ability to crystallize in both $P3_2$ and $R3$ (Supplementary Fig. 9). In these scenarios, $R3$ preferred low concentrations of divalent ions and organic solvents, and $P3_2$ required high salt (Supplementary Table 10) with drastically different lattices (Supplementary Fig. 10). When compared to the junctions that did not yield crystals in the 4 × 5 system, six (J11, 12, 13, 17, 18, and 27) junctions consistently proved fatal (Supplementary Table 11), and we suggest that these sequences could be considered unwise options for future design decisions.

The average cell constants for the $P3_2$ crystals were $a = b = 68.29$ Å $c = 55.68$ Å. The $c$-axis, in particular, was ~5 Å shorter, and had a much larger degree of variability ($\sigma = 1.97$), than those that crystallized using the 4 × 5 motif (Supplementary Table 12). The broad range of the short axis spanned from 52.77 to 60.36 Å, and we posit that the flexible axis lengths could be responsible for rendering a larger number of junctions fatal than the 4 × 5 system. In crystals containing $R3$ symmetry, the average cell for each was $a = b = 114.9$ Å $c = 49.77$ Å with the $c$-axis confined to a tighter regime ($\sigma = 0.75$). The cavity volumes were also markedly different from the $P3_2$ cavity volumes (~614.7 nm$^3$) compared with ~532.1 nm$^3$ in the more densely packed $R3$ structure (see Supplementary Fig. 11 for calculation details). The average junction angles of 54.60° ($\sigma = 1.44$) and 58.37° ($\sigma = 2.3$) for the $P3_2$ and $R3$ symmetries, respectively (Supplementary Table 13), along with their preference for salts or organic solvents, were likely the major contributing factors that yielded the divergent lattices.

**Role of the junction flanking sequence on symmetry in the 4 × 6 motif.** Because the 4 × 6 system yielded $R3$ symmetry in 5 of 17 junctions, we considered whether the downstream sequences adjacent to the junction could play a role in crystallization efficiency, junction angle, and symmetry preference, or if the HJ alone was the singular determinant. To investigate this possibility, we designed "scrambled" sequences with targeted base substitutions (Supplementary Fig. 12), while maintaining GC content, along each "stem" (Fig. 3a, b). Contrary to the buffer preference observed with the native $P3_2$ crystals, the scrambled systems exhibited an exclusive preference towards low salt buffers (Supplementary Fig. 13 and Supplementary Table 14), much like the native $R3$ crystals. Remarkably, only J1 and J2 retained the $P3_2$ symmetry, with all others yielding an $R3$ lattice (Supplementary Table 15). It is also noteworthy that J1 and J2 ($P3_2$), along with the $R3$ systems, also shared a preference for low salt conditions, unlike the original $P3_2$ crystals. This change in buffer preference suggests that global sequence content can indeed influence self-assembly behavior. Further, we observed modest improvements in resolution, reaching as high as 2.7 Å in J36. Notably, all junctions preventing crystallization compared to its original counterpart remained fatal (Supplementary Table 16) and all relative cavity dimensions and volumes remained unperturbed (Supplementary Figs. 14 and 15).

The role of longer-range sequence effects on junction angles remains a largely open question, but the effect in our crystal system is significant. There appeared to be only a marginal difference in the average cell lengths in the $R3$ scramble crystals ($a = b = 113.04$ $c = 51.10$; Supplementary Table 17) relative to the native 4 × 6 sequences (Supplementary Table 12): the $a$, $b$ axis trended ~2 Å shorter and $c$ ~1.3 Å longer, whereas in the J1 and J2 cases, the crystals were in good agreement with the original 4 × 6 motif. The average angles of the $R3$ and $P3_2$ crystals were 61.00° ($\sigma = 1.21$) and 58.05° ($\sigma = 1.39$), respectively. Although the average angle in the $R3$ crystals is nearly 3° higher than those from the native sequence (Supplementary Table 18), the scrambled sequence angle is calculated from a larger sample size than the smaller angle calculated in the native 4 × 6 structures (58.37°), with a significantly larger standard deviation (2.33). By contrast, the calculated angle for the $P3_2$ crystals was 58.05° ($n = 2$, $\sigma = 1.39$), compared with 54.60° ($n = 16$, $\sigma = 1.44$), a less accurate average due to a small sample size, so we posit that 54.60° more closely reflects the average observed angles in the 4 × 6 systems.

**Conserved ion binding sites within the junction structure influence crystallization.** We previously reported[15] the presence of arsenic ions at two opposing positions (Pos1 & 2, Supplementary Figs. 16 and 17) of the junction, due to the cacodylic acid contained in the crystallization buffer (Fig. 4). In a number of cases, we also observed a clustering of ions (Pos3; Supplementary Fig. 17) within the minor groove neighboring Pos2, but shared no apparent interactions with the junction. Pos1 and 2 were readily observable in the electron density maps at either one or both of these conserved sites in a significant number of the structures, with no apparent respect to motif or symmetry (Fig. 4). Although in the majority of cases, the individual junctions, regardless of design parameters, had a preference for crystallizing in cacodylic acid within the 6.0–6.5 pH regime, not all crystals were confined to this requirement. In the 4 × 5 system, J25 and J34, both displaying $P3_2$ symmetry, crystallized in 50 mM Tris pH = 8.0 buffer containing cobalt hexamine (CoH$_{18}$N$_6$) and 10 mM MgCl$_2$, and 50 mM HEPES pH = 7.5 with 20 mM MgCl$_2$, respectively. Only cobalt could account for the different peaks in the $F_o$–$F_c$ maps in J25, whereas in J34, and J22 and J23 in the 4 × 6 and

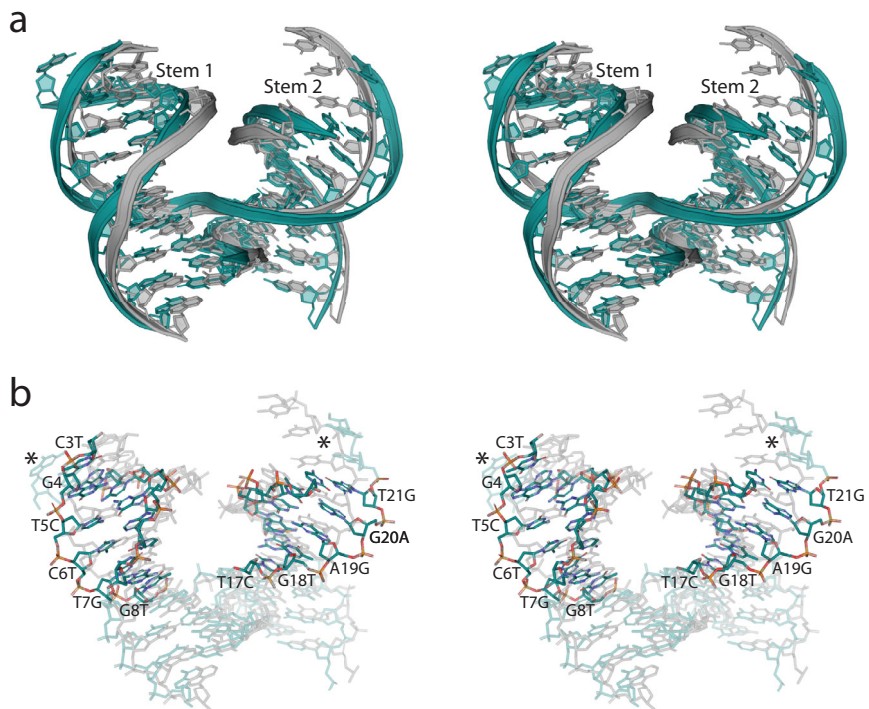

**Fig. 3 Stem sequence perturbations alter junction angles and influence global symmetry. a** Stereoview of a superposition of the J10 junction structure using the original 4 × 6 sequence motif with $P3_2$ symmetry (gray), with the scrambled sequence version containing $R3$ symmetry (teal). The modified sequences were located within the two downstream stem (1 & 2; indicated) regions containing the same GC content as the original 4 × 6 sequence version. The effect of the scrambled sequence on the geometry of the junction is visually apparent when comparing the superimposed Stem 1 & 2 regions. The dramatic difference in junction angle, and its influence on symmetry is evident (Supplementary Fig. 15). **b** Stereoview of a stick representation of the superimposed J10 structures in (**a**) with all base modification sites between the original and scrambled sequence version of the 4 × 6 J10 structures are indicated in Stems 1 & 2. Asterisks are included to provide attention towards the sticky end regions that significantly diverge as an apparent result of the angles induced by the modified stem sequences. Atoms are indicated using the following: carbon (teal), nitrogen (blue), oxygen (red), and phosphate (orange). All regions containing identical sequence are left translucent.

4 × 6 scramble systems, respectively, contained $Mg^{2+}$. The modeled ions were readily superimposable, and well-coordinated at Pos1 & 2 (Supplementary Fig. 18).

**Molecular dynamics simulations of individual HJs reveal interhelical angle differences.** Full-atomistic MD simulations were performed for all 36 immobile HJs to explore their dynamics in solution and compare the properties of the crystallizing and non-crystallizing junctions (see the "Methods" section in the Supplementary Information for full technical details). In over 224 µs of simulations, the force-field performance was deemed satisfactory, as stable base pairing and B-form helical topologies were observed in agreement with our crystallographic experiments and earlier studies[28,53–55]. The simulations revealed relatively minor differences in the interhelical dynamics among the 36 immobile HJs. We present the median angle values, as well as histograms of angle populations in Supplementary Table 19 and Supplementary Fig. 18, respectively. The interhelical angle rapidly fluctuated on the microsecond timescale, reflecting the greater conformational freedom of the HJs in solution. Its median value was typically lower than the one seen in the self-assembled DNA crystals reported in this work, likely reflecting a genuine influence of the respective environments (crystal lattice vs. free solution), as the simulation values are more consistent with those reported for X-ray structures of isolated HJs[56]. Still, for the majority of the HJs the difference was less than 5°. We note that the most unusual

interhelical angle values and distributions were observed for the J11 and J18 junctions, both of which never crystallized. The excessive interhelical dynamics and angle preferences incompatible with the lattice structure could be a factor contributing to the crystal growth inhibition for these two specific junctions.

**The ion binding ability of the junction could determine its potential to form 3D lattices.** The most significant difference between the individual junctions observed in MD simulations was their ability to form distinct potassium ion binding sites near the junction branching point. In all cases, the ion formed a bridge between the phosphate right at the branching point and one or two closest bases. These sites were in good agreement with the Pos1 and Pos2 sites observed in the experimental crystal structures (Figs. 4 and 5a). Since the base atoms of the branching point base pairs were involved in the ion coordination, the ion-binding sites were highly different among the 36 immobile HJs. The most obvious and striking difference was that the HJs which never crystallized in our experiments (J11, 12, 13, 18, and 27) were also those which consistently showed no ability to form these specific ion-binding sites in simulations. J17, which also never crystallized, did so in a negligible portion (0.02%) of all simulation frames (Fig. 5b). All the other HJs both crystallized in our experiments and formed these ion binding sites in our simulations to some degree (Fig. 5b). The mean incidence of binding in the non-fatal junctions was 0.53 ($\sigma = 0.28$), suggesting that the

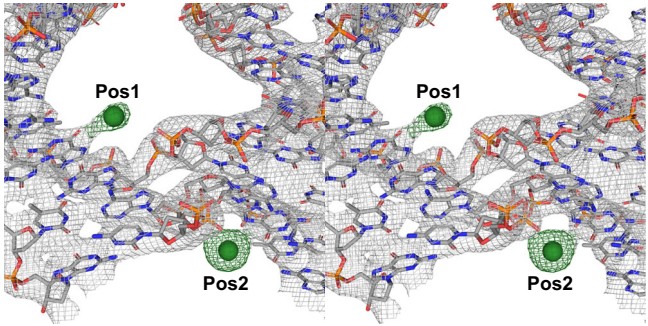

**Fig. 4 The junction crossovers contain unique ion binding positions.** Stereoscopic view, using J21 in the 4 × 5 system, $2F_o$–$F_c$ electron density accounting for the bases at the crossover regions are contoured at $\sigma = 2.0$, and the individual ion positions 1 and 2 (indicated Pos1 and Pos2) are independently contoured in the corresponding electron density at $\sigma = 4.0$. The presence of arsenic at these sites was substantiated by transferring the crystals into TAE-$Mg^{2+}$ (40 mM Tris, 20 mM acetate, and 1 mM EDTA pH = 8.6), and subsequently freezing the crystals. The crystals were scanned at the arsenic K edge ($\lambda = 1.04$ Å) where the corresponding arsenate peak was present. No other components within the crystallization buffers could account for the resulting peaks in the $F_o$–$F_c$ difference maps for the ions at their corresponding sites. Atoms are indicated using the following: carbon (gray), nitrogen (blue), oxygen (red), phosphate (orange), and arsenic ions (green spheres).

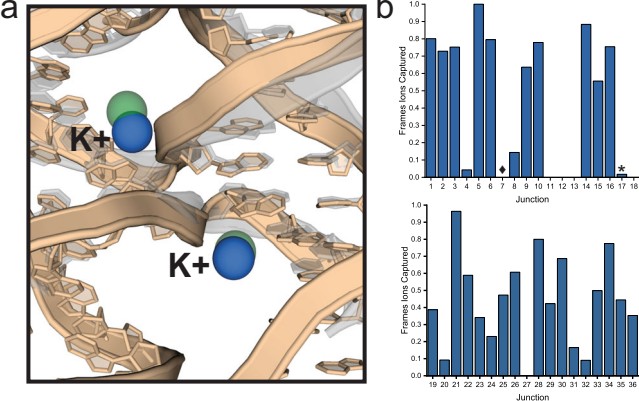

**Fig. 5 MD simulations relate capture of ions with lattice formation.** **a** Superimposed structures of the 4 × 5 J10 crystal structure (translucent gray) with a snapshot from the J10 simulation in solution (tan). The arsenic binding sites (green spheres) in the crystal structure near the branching point overlap with spontaneously formed potassium (blue spheres) binding sites in the solution structure simulations. **b** Graphs showing the incidence of ion capture near the branching point in simulations of all 36 junction sequences. The consensus "fatal" junctions (J11, 12, 13, 17, 18, and 27) show no ability for ion capture with the exception of J17 (asterisk) which did so to a negligible degree compared to the crystallizing junctions. All other junctions resulting in crystals demonstrated the ability for ion capture to a significant degree, with only a single outlier (J7, diamond). J7 robustly crystallized, but showed no ability to capture ions in both experiments and simulations, suggesting the ion binding is not essential for crystallization of this single junction.

ability to capture ions is pivotal to the ability to crystallize the immobile HJs. We speculate that the branching-point ion-binding site could be stabilizing the DNA strand exchange during the formation of the lattice and thus facilitate the crystal growth. This hypothesis is supported by the fact that the junctions unable to form this ion-binding site, or those which do so to a lesser degree,

are consequently among those that either do not grow crystals at all, or that are more sensitive to the crystallization conditions. The sole exception was J7 which, despite possessing an appropriate branching point sequence to do so, did not form the ion-binding site in simulations or experiments, but was still able to crystallize.

Initially, there might be an apparent contradiction associated with the experimentally detected presence of arsenic from a cacodylate anion in the region where the simulations localize $K^+$ cations (Fig. 5). However, we suggest that these observations can be reconciled. To our knowledge, all experimental structures of HJs in the PDB database reported by diverse independent groups suggest that cations (e.g., $Na^+$, $Mg^{2+}$, $Sr^{2+}$, $Ba^{2+}$) should bind at this location. This result is consistent with the highly negative molecular interaction potential (Supplementary Fig. 20) calculated here, which is a hallmark of cation-binding sites. However, in the majority of the crystal structures reported here, the cacodylate anion indeed fits best into the electron density map of our X-ray structures, due to the presence of sodium cacodylate in the crystallization solution (Supplementary Fig. 21). No other buffer component could form any reasonable contact distances at this location with the junction itself. A complete description of the comprehensive, experimental evidence for this arsenic has been described in our previous work[15]. This seemingly counter-intuitive result can be rationalized in several ways: First, the cacodylate could be stabilized at this site by hydrogen bonding and interactions with the solvent, both of which are known to provide stabilization for anionic binding even at regions with negative surface potentials[57,58]. Second, there could be one or more sodium counterions associated (catenated) with cacodylate at this site. Cacodylate anions are known to form interactions with negatively charged segments of nucleic acids in this fashion[59,60]. Lastly, there could be multiple $Na^+$ ions (the counterion of cacodylate) present around the anion. Due to the nominal resolution of our structures, which is in the ~3 Å range, it is not possible to unambiguously identify all the interacting species, nor is it possible to fully describe the exact coordination, such as the involvement of water molecules and $Na^+$. The binding of $Na^+$ could compensate or even over-compensate the cacodylate negative charge, effectively re-creating the common cation-binding site seen in simulations and other experimental structures. The $Na^+$ ions are often fluctuating and their coordination requirements are pliable, rendering them entirely invisible to us. Furthermore, as suggested by simulations, there could be significant local dynamics that obscure the densities with averaging. Past MD simulations reported highly variable $Na^+$ binding sites, from stiff to dynamic ones, around nucleic acids[61], and it is entirely possible that invisible $Na^+$ cations can bridge the cacodylate anion with the DNA carbonyl and phosphate groups. While the inability to describe the interaction site in greater detail is a limitation of the current work, it is abundantly clear that accommodation of ions at this location is a strict requirement for the junction to stabilize, and to effectively crystallize.

In this work, we have reported a systematic study of all 36 immobile HJ sequences across two different crystal systems, findings that can, in principle, be extended to any 3D DNA crystal and potentially other types of DNA architectures. We show that J1 (or any other junction) should not be considered a privileged option for designing self-assembled lattices. Rather, a host of other sequence combinations discussed herein should be explored, with many of them providing potentially superior performance. We make the essential observation that several junctions—including J11, 12, 13, 17, 18, and 27—are universally fatal to crystallization and should be avoided in future crystal designs, unless perhaps alternate isomers are considered. Another major observation is the importance of stem region sequences

outside the HJ, given that this sequence can control crystal symmetry and lattice architecture, and improve resolution in non-obvious ways. We also elucidated the key role of ion coordination in driving these effects. Scaling up DNA nanostructures to larger assemblies—for DNA crystals, as well as 1D and 2D lattices, and potentially DNA origami systems—will likely require taking into account sequence-dependent effects both at the level of stem and HJ geometries. Finally, the angle distributions obtained from our MD studies of junctions can improve accuracy of coarse-grained models and DNA nanotechnology design tools to more accurately represent nanostructures.

In summary, our experimental results are corroborated by the largest scale MD (224-μs-aggregate) simulation performed to date on HJs. The simulations elucidated the role of sequence-dependent flexibility and solvent effects on HJ conformations. This work also demonstrates that HJ sequence plays a non-trivial role in the ability of DNA crystals to form, and can dramatically influence crystal symmetry. Furthermore, we have provided a systematic and comprehensive sequence-structure study on self-assembled DNA crystal systems using all 36 junctions. Lastly, the study unexpectedly revealed specific molecular interactions (ion binding) using both experimental and modeling parameters, that has never been performed on this scale.

## Methods

**Crystallization and data collection**. All oligonucleotides were purchased from Integrated DNA Technologies (Coralville, Iowa) and purified using 14% denaturing polyacrylamide gel electrophoresis (PAGE) or HPLC. Following purification, the pelleted DNA was resuspended in nanopure $H_2O$ and washed 5× using 3 kDa molecular weight cut off filters (Amicon) to remove any remaining salt. A stock of the three-component strands (S1:S2:S3) was made with final concentrations of 30:120:120 μM for each construct. The sitting drop vapor diffusion method was performed with Cryschem plates (Hampton Research) using an adaptation of a discontinued DNA crystallization screen from a commercial vendor (Sigma Aldrich) containing a sparse matrix of 48 conditions (Supplementary Table 3). 500 μL of each corresponding condition was added to each reservoir, and a total drop volume of 6 μL was prepared containing a mixture of a 2:1 ratio of DNA stock to the corresponding reservoir solution with a final drop concentration of 30 μM. For junctions where crystallization was challenging, multiple rounds of screening using the 48-condition sparse matrix screen (Supplementary Table 3) were performed by varying conditions such as DNA concentration, buffer pH, salt concentration, and annealing times to allow for a reliable determination of the efficacy of the junction. The plates were then placed in a chilling incubator (Torrey Pines Scientific, Carlsbad, CA), and allowed to equilibrate to 60 °C for 1 h and then cooled using a linear gradient to 25 °C at a rate of 0.3 °C/h. The resulting crystals were imaged using a light microscope (Supplementary Figs. 6, 9, 10), and then cryo-protected using an artificial mother supplemented with 30% glycerol by direct addition to the drop. The crystals were subsequently harvested using cryo-loops (Hampton Research) and cryo-cooled by immediate submersion into a liquid nitrogen bath. All data were collected in a nitrogen cold-stream (100 K) at the corresponding beamlines indicated in Supplementary Table 1.

**Data processing and structure solution**. All diffraction data were processed in HKL2000[62], and the initial phases were calculated using molecular replacement in Phaser[63] from the PHENIX[64] suite of programs with either the J1 4 × 5 structures 5KEK and 6X8C, as the initial search models for the duplex and junction structures, respectively, and the J1 4 × 6 structures 5VY6 and 6XNA for the duplex and junction structures containing $P3_2$ symmetry in that system, respectively. However, with few exceptions, the majority of the 4 × 6 $R3$ symmetry crystals required another $R3$ model of one of its counterparts as the ideal search model. Multiple rounds of model building were performed in Coot, with the initial model first treated as a single rigid body, followed by subsequent iterative rounds using restrained refinement in REFMAC[65] from CCP4[66], along with real space, and XYZ coordinate calculation in phenix.refine. All ions were modeled into regions of $F_o–F_c$ difference density with a contour level $\geq \sigma = 3.0$ and refined. Atom occupancies and B-factor calculations were then refined, along with simulated annealing to conclude refinement. All refined models used an $R_{free}$ set containing 5–10% of the unique reflections for each structure. The coordinates and structure factors, totaling 134 unique structures, were deposited in the Protein Data Bank (PDB), and the corresponding accession codes are listed in Supplementary Table 20. Data collection and refinement statistics are all summarized in Supplementary Table 1 which is divided according to their respective crystal systems. All main text and Supplementary Information structure figures contained in this report were prepared using PyMOL[67].

**Molecular dynamics simulations**. We have used the structure of the junction J1 crystallized in 4 × 5 lattice geometry (PDB:5KEK)[15] as the starting structure for all simulations. Each arm of the HJ was extended to contain at least eight base pairs, with the extended parts of the helices taken from the adjacent cells of the crystal lattice. The resulting construct contained 64 nucleotides and was used as the initial structure for simulations of all the 36 immobile junction sequences, by substituting the branching point base pairs accordingly. We have used xLeap module of AMBER18[68] to prepare the topology and coordinate files. The latest AMBER OL15 DNA force field[69] was used in all reported simulations. During the revision, we also tested the alternative parmbsc1 DNA force field[70] for a subset of the HJs; however, with bsc1 we detected a significant population of non-native and possibly spurious β/γ g + /t backbone conformations (see Supplementary Fig. 22 and the accompanying text (Supplementary Discussion 1). This observation is also consistent with recent reports[71]. We thus suggest that for the present system the OL15 force field might be the optimal choice. Each HJ structure was solvated in an octahedral box of SPC/E water molecules[72] with a minimal distance of 16 Å between the solute and the box border (Supplementary Fig. 23). The 0.15 M salt concentration was established by the addition of KCl ions[73]. The relative ion positions were then compared to the ion sites in the crystal structures. Although the MD conditions and description of ion binding are not identical to the experimental conditions, the simulations should quite realistically reflect the relative overall propensities of different sequences to form the ion-binding site. Next, we performed pre-production equilibration and minimization of each system. The first minimization was performed with positional restraint of 25 kcal/mol/Å² placed on the DNA followed by an equilibration run using the same positional restraint in which the system was heated from 100 to 300 K on the timescale of 10 ps. This was then followed by a series of six minimizations and equilibration runs with 5, 4, 3, 2, 1, and finally, 0.5 kcal/mol/Å² positional restraint placed on the DNA. Each minimization consisted of 500 steps using the steepest descent method followed by 500 steps using the conjugate gradient method. Each equilibration, except the first one, was performed for 5 ps. For all junctions other than J1, an extra minimization step was taken at the beginning to optimize the initial geometry of the branching point base pairs. Production simulations were performed with the pmemd.cuda[74], using periodic boundary conditions and NPT ensemble, and the standard simulation protocol[75]. The length of each simulation was 1 μs and four independent simulations were performed for all 36 junctions. Simulations of selected HJs were then extended up to 20 μs to verify the convergence (Supplementary Fig. 24a, b). The analyses were performed with cpptraj and VMD[76,77], using the combined simulation ensemble of each individual junction. The interhelical angles of the junctions in simulations were measured as $J_{twist}$ parameters according to the definition previously described by Watson et al.[56] where the two helical axes of the stacked helical arms of the junction are represented by a vector and the $J_{twist}$ calculated as their dot angle product. Note that our simulations sampled transitions between right-handed and left-handed junctions which, however, have the same $J_{twist}$ values assigned by the Watson et al. 2004 definition. To differentiate the handedness, we defined an additional vector perpendicular to the junction branching point and then used it to calculate the dot angle product with the cross-product vector of the two vectors representing the helical axes. The value of this second dot angle was then used to differentiate the handedness of the junction with values above and below 90° corresponding to right-handed and left-handed junctions, respectively. The $J_{twist}$ values of the left-handed structures were subsequently normalized by −1.

**Reporting summary**. Further information on research design is available in the Nature Research Reporting Summary linked to this article.

## Data availability

All coordinates and structure factors generated in this study have been deposited in the RCSB Protein Data Base under the following accession codes (4 × 5 duplex structures: J1-5KEK, J3-6WQG, J5-6WRB, J6-6X8B, J7-6WSN, J8-6WSO, J9-6WSP, J10-6WSQ, J14-6WSR, J15-6WSS, J16-6WST, J19-6WSU, J20-6WSV, J21-6WSW, J22-6WSX, J23-6WSY, J24-6WSZ, J25-6WT0, J26-6WRJ, J28-6WRI, J29-6WT1, J31-6WRC; 4 × 5 junction structures: J1-6X8C, J3-6XDV, J5-6XDW, J6-6XDX, J7-6XDY, J8-6XDZ, J9-6XEI, J10-6XEJ, J14-6XEK, J15-6XEL, J16-6XEM, J19-6XFC, J20-6XFD, J21-6XFE, J22-6XFF, J23-6XFG, J24-6XFW, J25-6XGM, J26-6XFX, J28-6XFY, J29-6XFZ, J31-6XG0, J32-6XGJ, J33-6XGN, J34-6XGO, J35-6XGK, J36-6XGL; 4×6 duplex structures with $P3_2$ symmetry: J1-5VY6, J2-7JPB, J5-7JPA, J7-7JPC, J8-7JP9, J10-7JP8, J16-7JP7, J20-7JP6, J22-7JP5, J23-7JON, J24-7JOL, J26-7JOK, J28-7JOJ, J30-7JOI, J31-7JOH, J33-7JOG; 4 × 6 junction structures with $P3_2$ symmetry: J1-6XNA, J2-7JFT, J5-7JFU, J7-7JFV, J8-6XO5, J10-7JFW, J16-7JFX, J20-7JH8, J22-7JH9, J23-7JHA, J24-7JHB, J26-7JHC, J28-6XO6, j30-6XO7, J31-6XO8, J33-6XO9; 4×6 duplex structures with $R3$ symmetry: J4-7JRY, J5-7JRZ, J31-7JS0, J33-7JS1, J36-7JS2; 4 × 6 junction structures with $R3$ symmetry: J4-7JHR, J5-7JHS, J31-7JHT, J33-7JHU, J36-7JHV; 4×6 scramble duplex structures: J1-7JKD, J2-7JKE, J3-7JKG, J5-7JKH, J7-7JKI, J8-7JKJ, J10-7JKK, J14-7JL9, J16-7JLA, J19-7JLB, J21-7JLC, J22-7JLD, J23-7JLE, J24-7JLF, J26-7JNJ, J30-7JSB, J31-7JSC, J33-7JNK, J36-7JNL, J36-7JNM; 4 × 6 scramble junction structures: J1-7JK0, J2-7JJZ, J3-7JJY, J5-7JJX, J7-7JJW, J8-7JJ6, J10-7JJ5, J14-7JJ4, J16-7JJ3, J19-7JJ2, J21-7JIQ, J22-7JIP, J23-7JIO, J24-7JIN, J26-7JIM, J30-7JI9, J31-7JI8, J33-7JI7, J34-7JI6, J36-7JI5). In addition, all corresponding

accession codes can be found in Supplementary Table 20. The input and output files for MD simulation data generated in this study have been deposited in the Zenodo database under accession code: 6381939. The raw MD simulation trajectory data are available upon request due to the large size of the datasets.

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

## Acknowledgements

Results shown in this report are derived from work performed at the Argonne Photon Source (APS), Advanced Light Source (ALS), and the National Synchrotron Light Source II (NSLS-II). The ANL Structural Biology Center (SBC) at the Advanced Photon Source (SBC-CAT) is operated by UChicago Argonne, LLC, for the U.S. Department of Energy (DOE), Office of Biological and Environmental Research under contract DE-AC02-06CH11357. The Berkeley Center for Structural Biology is supported in part by the Howard Hughes Medical Institute. The ALS is a DOE Office of Science User Facility under Contract No. DE-AC02-05CH11231. The ALS-ENABLE beamlines are supported in part by the NIH, National Institute of General Medical Sciences (NIGMS), grant P30 GM124169. Results from beamlines AMX (17-ID) and FMX (17-BM) at the NSLS-II, which is a DOE Office of Science User Facility operated for the DOE Office of Science by Brookhaven National Laboratory under Contract No. DE-SC0012704. The Life Science Biomedical Technology Research resource is primarily supported by the NIH, NIGMS through a Biomedical Technology Research Resource P41 grant (P41GM111244), National Science Foundation Division of Materials Research (NSF2004250), and by the DOE Office of Biological and Environmental Research (KP1605010). This work was supported in part with projects SYMBIT reg. number CZ.02.1.01/0.0/0.0/15_003/0000477 financed by the ERDF (M.K. and J.S.) and 21-23718S by the Czech Science Foundation (M.K. and J.S.). N.S. acknowledges startup funds from Arizona State University. H.Y., N.S., and P.S. gratefully acknowledge support from the National Science Foundation Division of Materials Research (NSF2004250). H.Y. was additionally supported by the Presidential Strategic Initiative Fund from Arizona State University.

## Author contributions

C.R.S. and H.Y. conceived of the project. T.M., A.B., and I.F. purified all DNA and prepared all crystals used in the work under the direction of C.R.S. and N.S., and C.R.S. performed all crystallographic data analysis and structure solution and refinement. M.K., J.S., and P.S. conceived of all molecular dynamics simulation experiments, and the analysis of the simulation data was performed by M.K. The experimental analysis of all junction angles was performed by M.M. under the direction of P.S. All authors discussed the results and provided manuscript feedback, C.R.S. and T.M. prepared all figures, and C.R.S., M.K., N.S., and H.Y. wrote the paper.

## Competing interests

The authors declare no competing interests.
