## [Peer Review File · Nature Communications]

REVIEWER COMMENTS

Reviewer #1 (Remarks to the Author):

I have read with interest the article titled: "Engineering Three-dimensional DNA Crystals Dictated by Immobile Holliday Junction Sequence and Molecular Dynamics" that Simmons et al. have submitted for publication in Nature Communications.

Because of my expertise in molecular simulations, I mostly focused in my review on the simulation component of the work. The authors employed fully-atomistic MD simulations of the 36 immobile HJs studied to explore their dynamics in solution and compare properties of the crystallizing and non-crystallizing junctions. The simulations lasted for about 144 μ s, and the force-field was validated by checking the formation of stable base pairing and B-form helical topologies in agreement with the crystallographic experiments. Despite the authors' claim that significant differences were recorded among the 36 immobile HJs as far as fluctuations in the interhelical angle are concerned, for the majority of the HJs the difference was less than 5°. I would not take this difference seriously. However, the finding that the most unusual interhelical angle values and distributions were observed for the J11 and J18 junctions, both of which never crystallized, is important. And as the authors note, the excessive interhelical dynamics and angle preferences incompatible with the lattice structure could be among the factors inhibiting the crystal growth for these specific junctions.

Overall, the simulation work seems nice and quite interesting, but there is a couple of points that the authors should clarify:

1) I noticed that all the simulations were performed with the AMBER OL15 DNA force field. What about the most recent parmbsc1 modifications of the AMBER ff specifically developed for DNA? PARMBSC1 is believed to constitute a general-purpose force field for DNA atomistic simulation, which has been tested extensively for nearly 100 systems. It results in MD trajectories that sample stable structures that remain close to experimental ones, while preserving hydrogen bonds and helical characteristics even at the terminal base pairs, where artifacts have been common with other force fields. I suggest that the authors repeat some of their simulations with the HJs that never crystallized in their experiments (J11, 12, 13, 18 and 27), and which did not show any ability to form the specific potassium ion binding sites near the junction branching point in the course of the MD simulations, with this ff to make sure that their finding is robust and not an artifact of the force field.

2) Running a few MD simulations with PARMBSC1 would also confirm the robustness of the MD results for the differences in the fluctuations of the interhelical dihedral angle among the various HJs.

3) It would be nice if the authors reported snapshots of the entire simulation cell for some of the HJs simulated.

4) An important problem in simulations of crystal structures is that of ergodicity. How can we be sure that the 144 microseconds are long enough to capture the tendency of the system to escape from the initially imposed crystalline structure and sluggishly move to a new one or even develop disordered domains? I guess this is a hard question, which means that we have to take the results of the simulations

with caution. In this case, repeating the simulations with another ff or with a larger system might help.

Reviewer #2 (Remarks to the Author):

In this paper, the Authors present 36 immobile Holliday Junctions sequences and extensively characterize the systems through crystallography and modeling. Most of the sequences explored were crystallizable and the ones universally fatal pointed out; sequence effects, symmetry, and ion roles were thoroughly elucidated.

I am not aware of any other report with such a complete study but despite the completeness of the approach, the relevance of the information obtained is not completely unveiled through the text. To this end, I strongly suggest better stating the limitations of the most recent works and relating them with this paper achievements. The impact of the work should be better addressed and put in more evidence, with a particular focus in the conclusion section.

I would recommend publication in Nature Communication after these minor revisions.

Reviewer #3 (Remarks to the Author):

This paper presents an impressive collection of both crystallographic and MD simulation data related to the structures of a large array of Holliday junctions. The authors analyzed the differences between the various sequences and associated crystal forms they obtained. Specifically, they tried to rationalize their findings through the analysis of ion binding sites in crystal structures and MD simulations. I see several issues related to this analysis and significant imprecisions in the text.

- Figure 4: the authors did not precise in the figure caption which four-way junction they used. They write about TAE-Mg²⁺ without giving further details on this. This should be better explained in the SI.
- Figures S16 and S17: Making one figure with these two would be interesting since it is difficult to grasp immediately the interest of having two separate figures. Labeling could be improved by writing under each panel what it refers too. More importantly, I couldn't find any discussion in the text explaining why these sites (Pos 1 and 2) are similar besides a very roughly similar spatial position. In fact, the authors suggest that Pos 1 and 2 could be similarly occupied by cacodylate ions (certainly an anion), Mg²⁺, Co(NH₃)₆,³⁺, all different ions with very different binding properties, volumes and charge. For instance, cacodylate is negatively charged and cobalt hexamine carries a +3 charge. Given this fact, the authors did not rationalize the binding positions of these ions, an important point to discuss
- Figure 5: The authors performed MD simulations with K⁺ as an ion and compare the positions of K⁺ with that of cacodylate of opposite charge as mentioned above. This is odd and needs thorough explanations especially in terms of specific binding site coordination. To me it is clear that the absence of such discussions makes the interpretations presented by the authors considerably speculative. Since the role of the ions in the stabilization of these Holliday junctions is one of the main aspects of this study, it looks like present data and the used methodology do not support the authors conclusions made

by the authors.

Minor point:

- Tables S3, S4, S9, S10 and S14: please indicate the Cacodylate co-ion. Cacodylate usually comes as a salt with Na⁺ or K⁺. Which one was used?

Reviewer #1

I have read with interest the article titled: “Engineering Three-dimensional DNA Crystals Dictated by Immobile Holliday Junction Sequence and Molecular Dynamics” that Simmons et al. have submitted for publication in Nature Communications.

Because of my expertise in molecular simulations, I mostly focused in my review on the simulation component of the work. The authors employed fully-atomistic MD simulations of the 36 immobile HJs studied to explore their dynamics in solution and compare properties of the crystallizing and non-crystallizing junctions. The simulations lasted for about 144 μ s, and the force-field was validated by checking the formation of stable base pairing and B-form helical topologies in agreement with the crystallographic experiments. Despite the authors’ claim that significant differences were recorded among the 36 immobile HJs as far as fluctuations in the interhelical angle are concerned, for the majority of the HJs the difference was less than 5°. I would n’t take this difference seriously.

However, the finding that the most unusual interhelical angle values and distributions were observed for the J11 and J18 junctions, both of which never crystallized, is important. And as the authors note, the excessive interhelical dynamics and angle preferences incompatible with the lattice structure could be among the factors inhibiting the crystal growth for these specific junctions.

Overall, the simulation work seems nice and quite interesting, but there is a couple of points that the authors should clarify:

We thank the Reviewer for their careful appraisal of our manuscript, and are gratified that they find the work interesting. We agree that the difference in interhelical dynamics is minor for most of the junctions. We specifically point out J11 and J18, which are both notable and significant outliers in this regard compared to all the crystallizing junctions. Therefore, we suggested their unusual interhelical dynamics could be contributing to the difficulties in their crystallization. However, we do not claim this to be the sole factor responsible for the non-crystallizing properties of these or the other junctions which never crystallized. In the revised version of the manuscript, we put additional emphasis on this interpretation. We also remove the misleading claim that *significant* interhelical angle differences were observed among the 36 immobile HJs.

1) I noticed that all the simulations were performed with the AMBER OL15 DNA force field. What about the most recent parmbsc1 modifications of the AMBER ff specifically developed for DNA? PARMBS1 is believed to constitute a general-purpose force field for DNA atomistic simulation, which has been tested extensively for nearly 100 systems. It results in MD trajectories that sample stable structures that remain close to experimental ones, while preserving hydrogen bonds and helical characteristics even at the terminal base pairs, where artifacts have been common with other force fields. I suggest that the authors repeat some of their simulations with the HJs that never crystallized in their experiments (J11, 12, 13, 18 and 27), and which did not show any ability to form the specific potassium ion binding sites near the junction branching point in the course of the MD simulations, with this ff to make sure that their finding is robust and not an artifact of the force field.

Based on the Reviewer’s suggestion, we have performed additional simulations of the J1, J5, J11 and J13 junctions using the parmbsc1 DNA force field. The Reviewer specifically recommended performing simulations of non-crystallizing junctions. However, we reasoned that simulating a selection of both universally crystallizing (J1 and J5) and non-crystallizing (J11 and J13) junctions could better underscore the observed differences, and actually provide a more solid answer for the question posed by the Reviewer regarding the force-field differences. The cumulative length of the additional parmbsc1 MD simulations was 16 μ s.

However, our analysis of the parmbsc1 trajectories revealed a significant problem with accumulation of spurious β/γ backbone conformations within the gauche+/trans region. Their cumulative occurrence within the entire HJ structure was ~5%, while there were significant differences among the individual dinucleotide steps, with some of them possessing this conformation in as much as 40% of the simulation time. The gauche+/trans conformation of the β/γ dihedrals is not supported by the experimental structures of the HJ, nor is it to our knowledge generally associated with any B-form DNA structures. Therefore, we suspect that it might be a simulation problem and possible force-field problem of the parmbsc1 force field. Among others, the problem occurred within the central regions of the HJs as well, which determine the key effects described in our paper, such as the interhelical dynamics and ion binding proclivity. Therefore, we do not consider these trajectories as optimal representation of HJ dynamics.

We note that the spurious gauche+/trans conformation of the β/γ dihedrals is entirely absent when using the OL15 force field. Therefore, we politely suggest that use of OL15 over parmbsc1, at least for the HJ structures, might in fact be a more appropriate choice. We of course do not claim that OL15 is a flawless force field.

In the Supporting Information of the revised manuscript we provide a brief explanation along with a Figure illustrating the gauche+/trans conformation of the β/γ dihedrals, together with a very short comment in the main text Methods.

We would like to point out the problems of bsc1 with the β/γ dihedrals have already been reported in the literature, including reports of distortion of B-DNA geometry by these substates (<https://doi.org/10.1021/acs.jctc.1c00682>). Their presence (without further analyses) was noted already in <https://pubs.acs.org/doi/abs/10.1021/acs.jctc.6b00186>. Therefore, it is not a specific observation related to our simulation protocol and we are consistent with some other groups. The problem appears to be amplified by the higher dynamics of the HJ system compared to B-DNA.

In summary, we politely suggest that OL15 is a justifiable choice for HJ simulations. In addition, OL15 has been released only few months before parmbsc1 (<https://doi.org/10.1021/acs.jctc.5b00716> and <https://doi.org/10.1038/nmeth.3658>). Therefore, parmbsc1 is not a newer force field but rather an independent version. In addition, OL15 might be a more complete reparametrization than bsc1 with respect to the original 1995 Cornell et al. AMBER force field (<https://doi.org/10.1021/ja00124a002>). OL15 is (including all the preceding versions) a complete reparametrization of all dihedrals compared to the Cornell et al., while bsc1 did not modify the beta potential, which to our opinion might somewhat affect its performance for some systems including the HJ junctions. We fully agree that bsc1 has been tested in the original work on an impressive set of systems, but some problems might have been overlooked in the extended dataset. For example, Z-DNA has been tested and suggested to be correct, while subsequent works demonstrated that for Z-DNA the bsc1 is inferior to bsc0, while OL15 is an improvement (<https://pubs.acs.org/doi/abs/10.1021/acs.jctc.6b00186>, <https://pubs.acs.org/doi/10.1021/acs.jctc.1c00697>). OL15 has also been widely used, and for example a recent benchmark study (<https://doi.org/10.1073/pnas.2021263118>) on sequence-dependent B-DNA elasticity has been based on OL15.

2) Running a few MD simulations with PARMBSC1 would also confirm the robustness of the MD results for the differences in the fluctuations of the interhelical dihedral angle among the various HJs.

In principle, we agree with the Reviewer that this could be a useful verification of the simulation dynamics. However, as noted in the answer above, we think that the unusual backbone substates with the use of parmbsc1 do not allow us to make such a comparison. We would indeed see differences between OL15 and bsc1, but these would be dominantly associated with the β/γ g+/t states sampled by bsc1. Therefore, to

confirm the robustness of our results, we instead significantly extended selected MD simulations using the OL15 force-field (see below).

3) It would be nice if the authors reported snapshots of the entire simulation cell for some of the HJs simulated.

We thank the Reviewer for this suggestion, and snapshots of the entire simulation cell and its nearest periodic space for selected HJ simulation are now reported as a new Supplementary Information Figure.

4) An important problem in simulations of crystal structures is that of ergodicity. How can we be sure that the 144 microseconds are long enough to capture the tendency of the system to escape from the initially imposed crystalline structure and sluggishly move to a new one or even develop disordered domains? I guess this is a hard question, which means that we have to take the results of the simulations with caution. In this case, repeating the simulations with another ff or with a larger system might help.

We absolutely agree with the Reviewer that cumulative length of our simulations likely does not represent ergodic sampling of the HJs' conformational space. We do not, for example, observe a single instance of junction opening or trans-isomerization, both of which are known to occur in solution structures of HJs. We suggest that for a system of this size (64 nucleotides and ~ 11,000 water molecules), reaching ergodicity would be exceptionally challenging even in the context of enhanced sampling simulations. Sampling of conformational space relatively close to the crystalline structure may in fact be advantageous to our study, as it allows us to observe in detail the dynamics and solvent interactions relevant to the conformational stages where the HJ transitions from the solution to crystalline environments. We do not think that sampling of, for example, unfolded or disordered states would necessarily provide greater accuracy for that comparison. Nevertheless, based on the Reviewer's comment, and to further support our conclusions, we performed extended MD simulations for junctions J1, J5, J11 and J13 using the OL15 force field. Each new simulation was run for 20 μ s (cumulative length 80 μ s), thus significantly increasing the sampling compared to the originally presented simulations. In these extended simulations, we observed HJ dynamics and behavior highly consistent with the shorter 1 μ s simulations. Therefore, although we admittedly do not achieve ergodicity, we suggest that our HJ simulations do in fact represent a sufficient sampling within the free-energy basin of interest, allowing us to formulate justified and plausible suggestions about the dynamics processes occurring within the folded HJ structures (such as the interhelical dynamics and ion binding). In the revised version of the manuscript, the analyses of the extended simulations and comparison with the shorter ones are included as a new Supplementary Information Figure and referenced in the main text.

Reviewer #2

In this paper, the Authors present 36 immobile Holliday Junctions sequences and extensively characterize the systems through crystallography and modeling. Most of the sequences explored were crystallizable and the ones universally fatal pointed out; sequence effects, symmetry, and ion roles were thoroughly elucidated.

I am not aware of any other report with such a complete study but despite the completeness of the approach, the relevance of the information obtained is not completely unveiled through the text. To this end, I strongly suggest better stating the limitations of the most recent works and relating them with this paper achievements. The impact of the work should be better addressed and put in more evidence, with a particular focus in the conclusion section.

I would recommend publication in Nature Communication after these minor revisions.

We thank the Reviewer for their kind comments on the originality and potential impact of our manuscript, and agree that we did not perhaps make a strong enough case for the advancements and achievements in the current manuscript. To this end, we added some additional material about the simulations early in the manuscript, and a new concluding paragraph in the main text (both additions are highlighted in green) to further strengthen the case for our work.

Reviewer #3

This paper presents an impressive collection of both crystallographic and MD simulation data related to the structures of a large array of Holliday junctions. The authors analyzed the differences between the various sequences and associated crystal forms they obtained. Specifically, they tried to rationalize their findings through the analysis of ion binding sites in crystal structures and MD simulations. I see several issues related to this analysis and significant imprecisions in the text.

We thank the Reviewer for their helpful suggestions, and have addressed them as outlined below.

• Figure 4: the authors did not precise in the figure caption which four-way junction they used. They write about TAE-Mg²⁺ without giving further details on this. This should be better explained in the SI.

We thank the Reviewer for pointing out this oversight, and have added the specific junction used, and the full components and pH of the TAE-Mg²⁺ buffer in the Figure 4 legend.

• Figures S16 and S17: Making one figure with these two would be interesting since it is difficult to grasp immediately the interest of having two separate figures. Labeling could be improved by writing under each panel what it refers to. More importantly, I couldn't find any discussion in the text explaining why these sites (Pos 1 and 2) are similar besides a very roughly similar spatial position. In fact, the authors suggest that Pos 1 and 2 could be similarly occupied by cacodylate ions (certainly an anion), Mg²⁺, Co(NH₃)₆³⁺, all different ions with very different binding properties, volumes and charge. For instance, cacodylate is negatively charged and cobalt hexamine carries a +3 charge. Given this fact, the authors did not rationalize the binding positions of these ions, an important point to discuss

In this case, we separated Supplementary Figs. 16 & 17 because we believe they serve two unique purposes. Supplementary Fig. 16 is to demonstrate that the relative regional clustering of ions at each position is maintained regardless of crystal packing (P3221, P32, or R3). This is to indicate that the presence of the ions at the conserved site serve to stabilize the junction itself, but likely do not explicitly dictate the symmetry alone. To further expand on the specific positions, we have provided a new additional Supplementary Fig. 17 with 2D topologies indicating the positions (Pos1 and Pos2) where they are specifically proximal to the bases that comprise the junction, and its flanking region. The expressed purpose for Supplementary Fig. 18 (previously 17), is to show a sampling of junctions for each system (4x5, 4x6, and 4x6 scramble), which not only preserve the binding region, but are consistently superimposable regardless of the coordinated species.

• Figure 5: The authors performed MD simulations with K⁺ as an ion and compare the positions of K⁺ with that of cacodylate of opposite charge as mentioned above. This is odd and needs thorough explanations especially in terms of specific binding site coordination. To me it is clear that the absence of such discussions makes the interpretations presented by the authors considerably speculative. Since the role of the ions in the stabilization of these Holliday junctions is one of the main aspects of this study, it looks like present data and the used methodology do not support the authors conclusions made by the authors.

We are grateful to Reviewer #3 for pointing out that we had not sufficiently describe the molecular origins of the ion coordination. In particular, we agree that it is important to clarify the ability of both Pos1 and Pos2 to coordinate both cationic and anionic species. We have provided a thorough description of how this is the case in the main text (highlighted in blue). In short, hydrogen bonding interactions, or additional stabilizing cations (e.g. Na^+) could be responsible for positioning the anionic arsenate ligand, although the resolution is not sufficient to unambiguously show this.

We have previously described the case for the arsenic containing anionic cacodylic acid molecules at each junction site. Specifically, the molecule yields distinct $F_o - F_c$ peaks attributed to the arsenic atom in the majority of the junction structures; however, we posit that at resolutions $\sim 3 \text{ \AA}$, the resolution is insufficient to reveal the remaining structural details of the cacodylic acid molecule, and therefore were only able to model the As ion in the apparent, existing electron difference density. During iterative rounds of refinement other buffer components from the crystallization mother liquor were placed into density to identify if any other atoms could provide sufficient scattering for the difference density evident in the structure; however, only arsenic could account for all residual $F_o - F_c$ peaks. Fluorescence scans at the arsenic absorption peak, were also used to confirm the presence of the ion.

Minor point:

• Tables S3, S4, S9, S10 and S14: please indicate the Cacodylate co-ion. Cacodylate usually comes as a salt with Na^+ or K^+ . Which one was used?

We thank the Reviewer for pointing out this oversight; sodium is the counterion in the buffer, and this has been added to each of the tables in the Supplementary Information.

REVIEWERS' COMMENTS

Reviewer #1 (Remarks to the Author):

In the revised version of the manuscript titled: "The influence of Holliday junction sequence and dynamics on DNA crystal self-assembly", the authors (C.R. Simmons, T. MacCulloch, M. Krepl, M. Matthies, A. Buchberger, I. Crawford, J. Šponer, P. Šulc, N. Stephanopoulos, H. Yan) have made indeed a good effort to address most of my comments on the original submission of their paper.

Most specifically, the authors went through the effort to implement the parmbsc1 force field in their simulations, as it is recommended for DNA simulations, and compare predictions with OL15. I liked it a lot. On the other hand, it is a pity that parmbsc1 produces spurious β/γ backbone conformations within the gauche+/trans region. As the authors explain, the gauche+/trans conformation of the β/γ dihedrals is not supported by the experimental structures of the HJ, and it is not associated with any B-form DNA structures.

Thus, I agree with them that there exists either a simulation or a parameterization problem with the parmbsc1 force field. Given other recent reports for the same problem concerning the β/γ dihedrals and the distortion of the B-DNA geometry, I would agree with the authors that OL15 is the safest choice. I also liked that the authors brought up this issue in their revised manuscript.

Overall, the revised manuscript presents a nice piece of work and, given that the authors have adequately addressed the rest of my comments, I am happy now to recommend their work for publication in Nature Communications.

Reviewer #2 (Remarks to the Author):

After the changes made and the answers received, I consider this work suitable to be published in Nature communications.

Reviewer #3 (Remarks to the Author):

The authors revised their manuscript.